# Effects of Obesity, Blood Pressure, and Blood Metabolic Biomarkers on Grey Matter Brain Healthcare Quotient: A Large Cohort Study of a Magnetic Resonance Imaging Brain Screening System in Japan

**DOI:** 10.3390/jcm11112973

**Published:** 2022-05-25

**Authors:** Keita Watanabe, Shingo Kakeda, Kiyotaka Nemoto, Keiichi Onoda, Shuhei Yamaguchi, Shotai Kobayashi, Yoshinori Yamakawa

**Affiliations:** 1Institution of Open Innovation, Kyoto University, Kyoto 606-8501, Japan; yamakawa@bi-lab.org; 2Department of Diagnostic Radiology, Hirosaki University Graduate School of Medicine Radiology, Aomori 036-8562, Japan; kakeda@med.uoeh-u.ac.jp; 3Division of Clinical Medicine, Department of Neuropsychiatry, Faculty of Medicine, University of Tsukuba, Tsukuba 305-8577, Japan; kiyotaka@nemotos.net; 4Department of Psychology, Otemon Gakuin University, Osaka 567-8502, Japan; onodak1@gmail.com; 5Department of Neurology, Shimane University, Izumo 690-0823, Japan; yamagu3n@med.shimane-u.ac.jp (S.Y.); kobayashishotai@gmail.com (S.K.); 6Department of Neurology, Shimane Prefectural Central Hospital, Izumo 693-0068, Japan; 7Institute of Innovative Research, Tokyo Institute of Technology, Tokyo 152-8550, Japan; 8Academic and Industrial Innovation, Kobe University, Kobe 657-8501, Japan; 9ImPACT Program of Council for Science, Technology, and Innovation, Cabinet Office, Tokyo 100-8914, Japan; 10Brain Impact General Incorporated Association, Kyoto 606-8501, Japan

**Keywords:** MRI, GM-BHQ, GM volume, BMI, blood biomarker, blood pressure

## Abstract

This study investigated the relationship between grey matter (GM) volume and blood biomarkers, blood pressure, and obesity. We aimed to elucidate lifestyle factors that promote GM volume loss. A total of 1799 participants underwent the brain dock as a medical checkup. Data regarding blood pressure, obesity measurements, and standard blood biomarkers were obtained. Further, brain magnetic resonance imaging (MRI), including high-resolution T1-weighted imaging, was performed. We calculated the grey matter brain healthcare quotient (GM-BHQ), which represents GM volume as a deviation value. After adjusting for confounding variables, multiple regression analysis revealed that body mass index (BMI) (*b* = −0.28, *p* < 0.001), gamma-glutamyltransferase (γ-GTP) (*b* = −0.01, *p* = 0.16), and fasting blood glucose (*b* = −0.02, *p* = 0.049) were significantly correlated with GM-BHQ. Although the current cross-sectional study cannot determine a cause-and-effect relationship, elevated BMI, γ-GTP, and fasting blood glucose could promote GM volume loss.

## 1. Introduction

Aging is a significant social factor, and understanding the factors contributing to healthy cognitive aging has become a public health priority [1]. Neuroimaging studies have revealed that grey matter (GM) volume gradually decreases after an individual’s 20s [2]. Thus, elucidating lifestyle factors that can help prevent age-related decreases in GM volume is especially important [3].

Lifestyle-related diseases such as obesity, diabetes mellitus, and hypertension can lead to a decrease in brain volume [4,5,6,7,8,9]. A large population-based study reported a relationship between reduced GM volume and cardiovascular risk scores calculated from diabetes diagnoses, a cholesterol-lowering medication, an anti-hypertensive medication, and waist-to-hip ratio [10]. However, most studies have focused on the diagnosis of diseases such as diabetes, hypertension, and hypercholesterolaemia. The relationships between brain volume and continuous variables such as fasting blood glucose, systolic blood pressure, and blood cholesterol have not been well investigated. Furthermore, it remains unclear which of the multiple factors associated with lifestyle-related diseases exert a particularly negative effect on GM volume.

To study GM volume decline, we used the grey matter brain healthcare quotient (GM-BHQ), which was approved as an international standard (H.861.1) by the International Telecommunication Union Telecommunication Standardization Sector (ITU-T). GM-BHQ represents the average value of the standard deviations of the volume of each region in the whole brain [11]. The advantage of the BHQ is that it makes it relatively easy to understand the state of the brain in terms of GM volume. Previous studies showed that GM-BHQ was correlated with age [11], fatigue [12], and dietary balance [13]. Furthermore, GM-BHQ was more strongly correlated with cognitive function than its regional subscales of hippocampus or parahippocampus, indicating that GM-BHQ reflects cognitive function better than hippocampal or parahippocampal volume [14].

In the present study, we investigated the relationship between GM-BHQ and blood pressure, obesity measures, and blood test data, including glucose tolerance, dyslipidaemia, liver function, and renal function. In addition, we aimed to elucidate lifestyle factors that could promote GM volume loss and brain aging.

## 2. Materials and Methods

### 2.1. Participants

Human experiments were carried out in accordance with the guidelines provided and approved by the Shimane University Institutional Committee on (20151028-1). All participants provided written informed consent to participate in the study, according to the Declaration of Helsinki.

Participants underwent the brain dock as a medical checkup between October 2013 and March 2019 at the Shimane Institute of Health Science. The brain dock includes MRI in addition to physical and laboratory examinations. The brain dock includes MRI and MR angiography of the brain, carotid ultrasound, and an assessment of mental status in addition to regular physical and laboratory examinations [15]. The aim of the brain dock was to assess health and detect brain diseases such as aneurysms, old infarctions, and bleeding. Participants underwent the brain dock at their discretion and expense. Therefore, participants tend to be interested in health.

Participants with gross MRI abnormalities such as large brain tumours and large infarcts or old haemorrhages were excluded. A total of 1799 participants were included. All participants were enrolled in an earlier published study [14], which analysed the relationship between cognitive decline and GM brain healthcare quotient (BHQ).

### 2.2. Data Collection

Data collected during face-to-face interviews of trained nurses included blood pressure; obesity measures of body mass index (BMI), waist circumference, and body fat percentage; and blood samples for standard biochemical assessments. Bodyweight, height, and fat percentage were obtained with a body composition analyser using bioimpedance. Waist circumferences were measured midway between the lowest rib and the iliac crest in the horizontal plane, with the patient at the end of a normal breath expiration. Blood pressure was measured once in the morning with a digital sphygmomanometer, using a suitably sized cuff. If the systolic blood pressure was 140 mmHg or higher, remeasurements were performed and the lower value was used. In addition, trained nurses obtained samples for blood count tests and blood chemistry in the morning after an 8–12 h overnight fast. For the blood chemistry test, samples were transferred to sterilized centrifuge tubes and allowed to clot at room temperature. Then, the blood samples were centrifuged for 10 min in a tabletop clinical centrifuge at 4000 rpm for serum separation. The items in the blood test were set by the facility according to the recommended items of the Japan brain dock society. The Japan brain dock society is an academic society for the prevention of stroke and dementia (http://jbds.jp, accessed on 1 April 2022). Clinical characteristics including age; sex; and the presence and medication of hypertension, diabetes mellitus, and hyperlipidaemia were recorded in person. Additionally, all participants underwent examination by a general physician.

### 2.3. MRI

MRI was changed from Siemens 1.5-T scanner to Philips 3.0-T scanner at the examination facility in January 2016 during the study period. The Siemens 1.5-T scanner was used for 1300 participants and the Philips 3.0-T scanner was used for 499 participants. Images were obtained with conventional T2-weighted, T1-weighted, fluid-attenuated inversion recovery, and T2* images. Additionally, three-dimensional T1-weighted images were acquired using magnetization prepared rapid acquisition with gradient echo (MPRAGE) and T1 turbo field echo (T1TFE) sequences. The acquisition parameters are shown in the Appendix A.

### 2.4. Image Processing for Brain Volume

The SPM12 software program (Statistical Parametric Mapping 12; Institute of Neurology, London, UK) was used to process images [16,17]. The 3D-T1WI in native space was segmented into GM, white matter (WM), and cerebrospinal fluid (CSF) images; spatially normalized to Montreal neurological institute (MNI) space; and modulated using the Diffeomorphic Anatomical Registration Through Exponential Lie Algebra (DARTEL) toolbox in SPM12 [16,18]. To preserve the GM volumes within each voxel, the images were modulated using Jacobian determinants derived from spatial normalization. The resulting modulated GM images were smoothed using an 8 mm full width at half maximum Gaussian kernel.

Proportional GM images were generated by dividing smoothed GM images by intracranial volume to control differences in whole-brain volume across participants. Intracranial volume was calculated by summing the GM, WM, and CSF images.

To evaluate the degree of brain volume, the proportional GM images were converted into a GM-BHQ [11], which is similar to the intelligence quotient. The mean value was defined as a BHQ of 100, and the standard deviation was defined as 15 BHQ points. Approximately 68% of the population is between BHQ 85 and BHQ 115, and 95% is between BHQ 70 and BHQ 130. GM-BHQ was calculated based on the database of Nemoto et al. [11], which contains the data of 144 healthy participants (64 women, 80 men; mean age = 48.4, SD = 8.1). First, 116 regional GM quotients were extracted for the 116 brain regions based on the Automated Anatomical Labelling atlas [19]. Second, we averaged the 116 regional GM quotients to produce GM-BHQ values.

### 2.5. Statistical Analyses

All statistical analyses were performed using IBM SPSS Statistics version 27 (IBM, Armonk, NY, USA).

First, the partial correlation was investigated between GM-BHQ and the following factors, classified into seven subgroups: (1) blood pressure: systolic blood pressure (SBP), diastolic blood pressure (DBP), and heart rate; (2) obesity measurements: BMI, waist circumference, and body fat percentage; (3) liver function: total protein, albumin, total bilirubin, aspartate aminotransferase (AST), alanine aminotransferase (ALT), and γ-glutamyltransferase (γ-GTP); (4) renal function: blood urea nitrogen (BUN), creatinine (Cr), and uric acid; (5) lipid metabolism: total cholesterol, triglyceride cholesterol, high-density lipoprotein (HDL) cholesterol, and low-density lipoprotein (LDL) cholesterol; (6) electrolytes: Na, K, Cl, and Ca; (7) glycometabolism: fasting blood glucose (Glu) and haemoglobin A1c (HbA1c); (8) blood cell measures: white blood cell (WBC) count, red blood cell (RBC) count, haemoglobin, haematocrit (Ht), platelet (PLT) count, and fibrinogen. Age, sex, and MRI machine were used as covariates. Patients on medications related to each subgroup were excluded (i.e., those on anti-hypertensive medication to analyse blood pressure). A flow diagram is shown in Figure 1. Bonferroni correction was applied for the correction of multiple comparisons. *p* < 0.002 was considered significant.

Second, multiple regression analysis models were used to examine the effect of the independent variables on the GM-BHQ. The factors with the highest correlation coefficient in each of the seven subgroups were selected if *p* < 0.05 in the partial correlation analysis. To select the variables using the results of partial correlation analysis as the primary outcome, a previous research method was referred [20]. Only one variable was selected in each subgroup because variables with similar meanings could cause multicollinearity (i.e., BMI and waist circumference in the obesity measurement subgroup). Patients taking medications for hypertension, hyperlipidaemia, or diabetes were excluded. *p* < 0.05 was considered to indicate statistical significance.

### 2.6. Data Availability

A minimal dataset is available on request from the corresponding author.

## 3. Results

### 3.1. Data of Participants

The mean age and standard deviation were 62.0 and 13.1 years old, respectively. The age range was 27–95. The participants’ data, including self-reported neurologic symptoms and MRI findings, are shown in Table 1. Blood biochemistry, blood pressure, and obesity data are summarized in Table 2.

### 3.2. Partial Correlation Analysis

The results of the partial correlation analysis are summarized in Table 2 and Figure 2. There were significant correlations between GM-BHQ and SBP (*r* = −0.10, *p* = 0.001), BMI (*r* = −0.19, *p* < 0.001), body fat percentage (*r* = −0.13, *p* < 0.001), AST (*r* = −0.11, *p* < 0.001), ALT (*r* = −0.12, *p* < 0.001), γ-GTP (*r* = −0.13, *p* < 0.001), Glu (*r* = −0.12, *p* < 0.001), and Alc (*r* = −0.09, *p* < 0.001).

Since the age range of the participants is large, we performed the same analysis for younger and older participants than the median as ad-hoc analyses (Appendix A). The numbers of the younger group (less than or equal to 64 years old) and older group (over 64 years old) were 891 and 908, respectively. BMI and γ-GTP showed significance for both the younger and older group. Waist circumference, body fat percentage, ALT, and Glu showed significance only for the younger group.

### 3.3. Multiple Regression Analysis

The multiple regression analysis included 813 participants because 631 individuals taking medications for hypertension, hyperlipidaemia, or diabetes and 355 individuals that did not have all the data for variables listed below were excluded. SBP, BMI, γ-GTP, uric acid, GLU, and Ht were selected as explanatory variables. These models also included sex, age, and MRI machine as additional covariates. The explanatory variables of BMI (*b* = −0.28, *p* < 0.001), γ-GTP (*b* = −0.01, *p* = 0.16), and Glu (*b* = −0.02, *p* = 0.049) significantly predicted GM-BHQ (Table 3 and Figure 2). BMI exhibited the highest standardization coefficient among blood biomarkers, blood pressure, and obesity measurements.

In the ad-hoc analyses for the younger and older group, BMI also showed significance and the highest standardization coefficient for both groups (Appendix A).

## 4. Discussion

The present results indicated that obesity measurements, blood pressure, and multiple blood biomarkers were significantly correlated with the GM-BHQ. Furthermore, although brain health is influenced by several factors, the current analysis indicated that BMI, γ-GTP, and Glu are related to smaller GM volumes.

In the multiple regression analysis, BMI exhibited a higher standardized partial regression coefficient than Glu or SBP. Furthermore, the same tendency was seen in ad-hoc analyses for younger and older participants than the median. Obesity not only leads to an increased incidence of lifestyle-related diseases [21], but also coincides with an increase in the risk of cognitive decline [22] and dementia [23]. Furthermore, brain MRI studies have revealed that obesity and high BMI are linked to decreased GM volume [4,5,6]. In obesity, inflammatory responses with subtle glial cell activation have been observed in the central nervous system and are referred to as neuroinflammation without peripheral immune cells [24]. Preclinical animal studies have also shown that high-fat and high-sugar diets are related to neuroinflammatory changes in the brain [25]. These previous studies suggest that obesity leads to a smaller GM volume. Interestingly, a recent study of 15,634 subjects in the UK Biobank showed that the effect of obesity was not mediated by blood pressure and blood biomarkers of glucose, lipids, and C-reactive protein [26]. In other words, obesity itself, rather than obesity-associated high blood pressure, diabetes, hyperlipidaemia, or inflammation in peripheral blood, has an effect on GM volume. Thus, preventing obesity can be especially important for brain health care and maintaining GM volume. On the other hand, the cross-sectional results in the present study could not determine a cause-and-effect relationship between smaller GM volumes and obesity. In addition, smaller GM volumes may cause obesity. The previous meta-analysis revealed that obesity is associated with reduced orbitofrontal cortex volume [27]. The orbitofrontal cortex is considered to play an important role in decision making [28] and neural circuits of feeding [29].

Furthermore, glycometabolism and blood pressure are associated with smaller GM volumes in healthy populations without medications, although a small number of participants were untreated. Interestingly, the present results indicate that reduced GM volume may also be associated with relatively high blood pressure or blood glucose levels that remain within the normal range. In other words, even among individuals without diabetes or hypertension, elevated blood glucose or blood pressure is associated with a smaller GM volume. There are consistent reports of associations between accelerated brain atrophy and patients with type 2 diabetes and hypertension [7,8,30]. A few studies have also suggested that prediabetes exerts adverse effects on brain health. Older adults aged over 64 years with prediabetes have a two-fold risk of incident Alzheimer’s disease [31]. In women, prediabetic status is associated with brain hypometabolism (as measured using FDG-PET) and poorer executive function and language performance, whereas these associations are not observed in men [32]. Although a previous study reported that prediabetes was not associated with hippocampal atrophy [32], high HbA1C was a risk factor for the progression of whole-brain atrophy, including GM and white matter volume, in a 6-year follow-up study of 201 participants [33]. Maillard et al. reported that elevated SBP was associated with reduced GM volumes in 579 healthy young adults (mean age: 39.2 years, SD: 8.4) recruited from a community-based cohort study [9]. Interestingly, Glu exhibited a higher standardization coefficient than SBP in the multiple regression analysis. Thus, Glu may be more strongly associated with GM volume loss than SBP.

LDL and HDL are important risk factors for cardiovascular disease, and lowering LDL and maintaining HDL is known to improve cardiovascular outcomes [34]. However, there was no significant correlation between GM volume and HDL or LDL levels in this study. This result is consistent with that of previous studies. A previous study of 789 older adults also reported no significant correlation between high HDL levels and brain volume [35]. Furthermore, Leeuw et al. reported that LDL and HDL were not associated with brain volume in 3962 participants [36]. Conversely, Leeuw et al. also reported that higher levels of small HDL in HDL subclasses were related to smaller brain volume [36]. The current knowledge on HDL subclasses is limited, although small HDL has been suggested to have more antioxidant and anti-inflammatory properties than lipid-rich large HDL [37]. Further studies are required to elucidate the relationship between brain health and lipid metabolism.

The current results indicate that γ-GTP, AST, and ALT are associated with a smaller GM volume. Furthermore, alcoholic liver damage and fatty liver are possible causes of elevated γ-GTP, AST, and ALT levels. Widespread brain atrophy in individuals with alcohol dependence has been consistently documented in MRI studies [38,39]. Additionally, recent studies suggest that even low to moderate alcohol consumption is a risk factor for brain atrophy [40,41], although the protective effect of a small amount of alcohol on cognition has also been reported [42]. However, individual differences in alcohol metabolism are well known [43]. Therefore, investigating whether low to moderate alcohol consumption without elevated γ-GTP, AST, and ALT levels is a risk factor for brain atrophy may be of interest to future researchers.

Our study has several strengths. The large sample size allowed us to confirm the result by additional analyses divided into two groups according to age. In addition, most of the past studies have been limited to one item [4,5,6,7,8,9], but in this study, various measurement items such as obesity, blood pressure, and blood glucose could be used for complex analysis. Furthermore, all participants were examined by general physicians and expert nurses.

There are also several limitations to this study. First, SBP values may have been unstable because the data were measured at a medical institution. Furthermore, it is possible that fasting before blood testing was inadequate in a few participants. Second, this was a cross-sectional study, and longitudinal changes in brain atrophy were not followed. Third, the participants in the brain dock system tended to be interested in health, which may have resulted in selection bias due to differences between the included participants and the general community-dwelling population. Fourth, the participants on medication were not evaluated due to a lack of detailed treatment status information. In particular, whether GM volume loss associated with diabetes and hypertension can be suppressed by controlling blood glucose levels and blood pressure is a critical issue.

In conclusion, smaller GM volume, which may be an important indicator of brain function, was positively related with following biomarkers: (1) systolic and diastolic blood pressure and heart rate; (2) body mass index, waist circumference, and body fat percentage, (3) blood biomarkers of liver function (AST, ALT, and γ-GTP), uric acid, glycometabolism (Glu and HbA1c), and Ht. Furthermore, the present results suggest that obesity is especially related to smaller GM volumes.

## Figures and Tables

**Figure 1 jcm-11-02973-f001:**
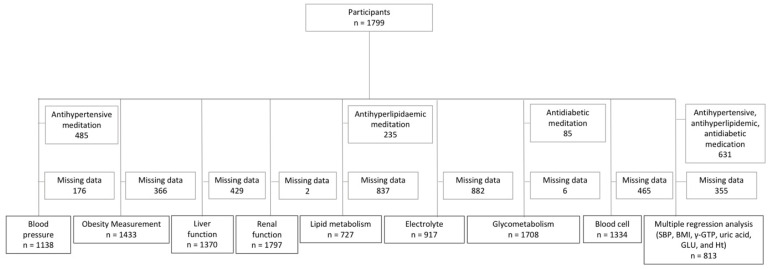
Flow diagram. This figure shows the flow diagram for the partial correlation analyses of seven subgroups and the multiple regression analysis. SBP—systolic blood pressure; BMI—Body Mass Index; γ-GTP—γ-glutamyltransferase; GLU—fasting blood glucose; Ht—hematocrit.

**Figure 2 jcm-11-02973-f002:**
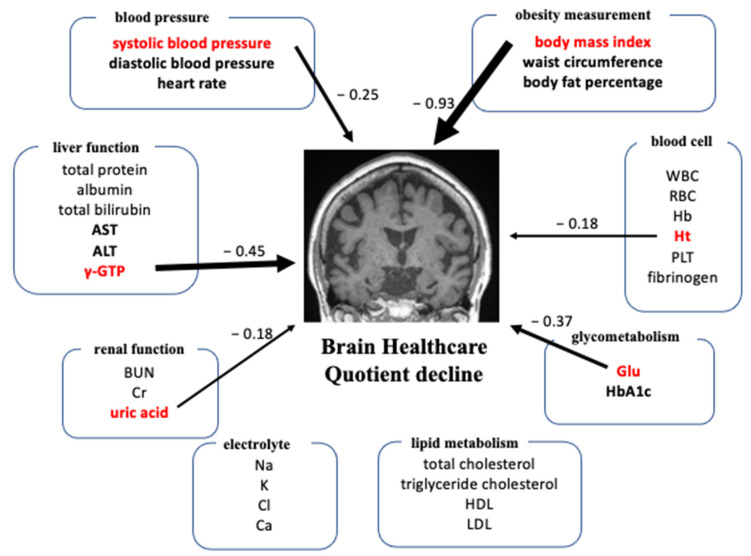
Illustration of the results of the partial correlation and multiple regression analysis. Bold black and red type denote the items showing significance in the partial correlation analysis without multiple comparison. The numbers and thickness of arrows show the standardization coefficient in the multiple regression analysis (dependent variable: GM-BHQ, explanatory variable: red bold).

**Table 1 jcm-11-02973-t001:** Demographic data for all participants.

Number of participants (female)	1799 (770)
Age, mean ± SD	62.0 ± 13.1
Self-reported neurologic symptom	
Dizziness	134 (69)
Headache	390 (223)
Tinnitus	330 (142)
Subjective cognitive decline	930 (455)
None of the above	588 (193)
No answer	27 (11)
Small infarction	218 (82)
Microhaemorrhage	236 (92)
Both small infarction and microhaemorrhage	67 (25)
Mini-Mental State Examination, mean ± SD	28.5 ± 2.2

The number in parentheses shows the number of females.

**Table 2 jcm-11-02973-t002:** Correlations between grey matter brain healthcare quotient and blood data, blood pressure, and obesity measurements.

	Mean ± Standard Deviation (Range)	Partial Correlation	*p*
**Blood pressure**			
systolic blood pressure	126.6 ± 16.8 (87–206) mmHg	−0.10	0.001 *
diastolic blood pressure	73.4 ± 10.9 (45–128) mmHg	−0.06	0.03
heart rate	63.1 ± 10.5 (37–117) bpm	−0.09	0.004
**Obesity measurements**			
body mass index	22.9 ± 3.2 (14.2–48.4)	−0.19	<0.001 *
waist circumference	82.6 ± 9.2 (55.5–127.6) cm	−0.07	0.009
body fat percentage	24.5 ± 6.4 (7.0–48.5) %	−0.13	<0.001 *
**Liver function**			
total protein	7.4 ± 0.4 (6.2–10.1) g/dL	0.00	0.921
albumin	4.4 ± 0.2 (3.4–5.5) g/dL	0.01	0.758
total bilirubin	0.8 ± 0.3 (0.2–3.5) mg/dL	0.01	0.672
aspartate aminotransferase (AST)	24.7 ± 11.5 (9–251) U/L	−0.11	<0.001 *
alanine aminotransferase (ALT)	23.2 ± 15.5 (3–215) U/L	−0.12	<0.001 *
γ-glutamyltransferase (γ-GTP)	41.0 ± 56.2 (7–1173) U/L	−0.13	<0.001 *
**Renal function**			
blood urea nitrogen (BUN)	15.1 ± 4.0 (5.6–55.1) mg/dL	0.00	0.984
creatinine (Cr)	0.8 ± 0.3 (0.4–8.7) mg/dL	−0.01	0.559
uric acid	5.3 ± 1.3 (0.7–11.1) mg/dL	−0.07	0.003
**Lipid metabolism**			
total cholesterol	211.0 ± 54.3 (24–2013) mg/dL	0.07	0.05
triglyceride cholesterol	109.9 ± 69.8 (28–924) mg/dL	−0.12	0.744
high-density lipoprotein (HDL)	64.4 ± 16.4 (30–155) mg/dL	−0.02	0.587
low-density lipoprotein (LDL)	120.0 ± 30.3 (15–236) mg/dL	0.05	0.147
**Electrolytes**			
Na	141.0 ± 1.8 (131–146) mEq/L	0.02	0.644
K	4.1 ± 0.3 (2.6–5.3) mEq/L	−0.03	0.45
Cl	102.7 ± 2.4 (95–111) mEq/L	0.00	0.979
Ca	9.4 ± 0.3 (7.6–11.0) mEq/L	0.00	0.903
**Glycometabolism**			
fasting blood glucose (Glu)	102.2 ± 19.3 (72–334) mg/dL	−0.12	<0.001 *
haemoglobin A1c (HbA1c)	5.6 ± 0.6 (3.7–10.9) %	−0.09	<0.001 *
**Blood cell values**			
white blood cell count (WBC)	54.8 ± 14.6 (18.4–169.3) × 10^2^/μL	0.01	0.792
red blood cell count (RBC)	460.5 ± 43.9 (241–600) × 10^4^/μL	−0.01	0.688
haemoglobin (Hb)	14.4 ± 1.4 (7.7–19.0) g/dL	−0.05	0.061
haematocrit (Ht)	42.3 ± 3.8 (24–55) %	−0.08	0.005
platelet count (PLT)	22.6 ± 6.1 (5–169) × 10^4^/μL	0.01	0.657
fibrinogen	288.0 ± 63.7 (140–630) mg/dL	0.06	0.056

* Statistical significance with Bonferroni correction for multiple comparisons.

**Table 3 jcm-11-02973-t003:** Multiple regression analysis for predicting grey matter brain healthcare quotient.

	*b*	Standard Error	Standardization Coefficient	*t*-Value	*p*
Systolic blood pressure	−0.01	0.01	−0.25	−0.13	0.194
Body mass index	−0.28	0.06	−0.93	−4.69	<0.001
γ-glutamyltransferase (γ-GTP)	−0.01	0	−0.45	−2.41	0.016
Uric acid	−0.13	0.15	−0.18	−0.02	0.397
Fasting blood glucose	−0.02	0.01	−0.37	−1.97	0.049
Haematocrit	−0.01	0.05	−0.18	−0.847	0.901

## Data Availability

Not applicable.

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
