# Peer review of "Effects of Obesity, Blood Pressure, and Blood Metabolic Biomarkers on Grey Matter Brain Healthcare Quotient: A Large Cohort Study of a Magnetic Resonance Imaging Brain Screening System in Japan"

_jcm, 2022, doi:10.3390/jcm11112973_

Round 1
Reviewer 1 Report
Please see the attached review.

Reviewer 2 Report
The paper described an investigation of the relationship between GM volume and various clinical measures. It is found that several parameters correlate with GM and that BMI is the most explanatory variable. The paper is well written, however there are several issues (described below) that would need addressing before publication.
- Were the subjects from the general populations or were they selective in some manner?
- It is not clear what the Brain Dock is, could you add a link or supplemental material with a list of the examinations undergone.
- In table one can you add the number of female respondents for each line as you have done with the Number of participants
- In the Data Collection section more references or details are needed, for example the Frankfort plane and the standard procedure for waist measurement
- Was blood pressure measured once or multiple times and was it acquired at the same time of day for each subject or randomly throughout the day?
- How many subjects were imaged with each of the scanners?
- To my recollection SPM produces tissue probability maps, how were these converted to Tissue masks?
- How was the intracranial volume measured?
- Does the DARTEL normalisation transform the images to the same space as the AAL atlas?
- I am unclear as the how the BHQ was calculated, what do you mean by the quotient from each region of the AAL?
- Following on I am not sure about the averaging, do you mean you take the 166 volumes and then simply average them? How does this differ from dividing the whole GM volume by 116?
- Why do you use the BHQ and not just the normalised GM volume?
- As you have volumes from the AAL segmentation did you repeated the analysis for each region to see if there were relationships with specific GM areas?
- Figure 1 seems to show move than the 7 groups of variables described in the text, in fact you seem to have 8 groups, two of which are labelled 7 in the text.
- In the discussion you state ‘reduced GM volume 192 may be caused by relatively high blood’ elsewhere you been careful to say associated with rather than caused so it should probably be changed here as well.
- The age range of the participants is large, although you have age as a covariate it would be interesting to assess if the same relationships are seen in, say, those older and those younger than the median.
- A further recent paper https://pubmed.ncbi.nlm.nih.gov/35145215/ should be included in the discussion
Round 2
Reviewer 1 Report
Please see the attached review.
